# Antifragile Control Systems: The Case of an Anti-Symmetric Network Model of the Tumor-Immune-Drug Interactions

Cristian Axenie [1,*], Daria Kurz [2,3] and Matteo Saveriano [4]

1    Audi Konfuzius-Institut Ingolstadt Laboratory, Technische Hochschule Ingolstadt, 85049 Ingolstadt, Germany
2    Fakultät für Gesundheit, Universität Witten/Herdecke, 58455 Witten, Germany
3    Interdisziplinäres Brustzentrum, Helios Klinikum München West, 81241 Munich, Germany
4    Department of Industrial Engineering, University of Trento, 38123 Trento, Italy
*    Correspondence: cristian.axenie@audi-konfuzius-institut-ingolstadt.de

**Abstract:** A therapy's outcome is determined by a tumor's response to treatment which, in turn, depends on multiple factors such as the severity of the disease and the strength of the patient's immune response. Gold standard cancer therapies are in most cases fragile when sought to break the ties to either tumor kill ratio or patient toxicity. Lately, research has shown that cancer therapy can be at its most robust when handling adaptive drug resistance and immune escape patterns developed by evolving tumors. This is due to the stochastic and volatile nature of the interactions, at the tumor environment level, tissue vasculature, and immune landscape, induced by drugs. Herein, we explore the path toward antifragile therapy control, that generates treatment schemes that are not fragile but go beyond robustness. More precisely, we describe the first instantiation of a control-theoretic method to make therapy schemes cope with the systemic variability in the tumor-immune-drug interactions and gain more tumor kills with less patient toxicity. Considering the anti-symmetric interactions within a model of the tumor-immune-drug network, we introduce the antifragile control framework that demonstrates promising results in simulation. We evaluate our control strategy against state-of-the-art therapy schemes in various experiments and discuss the insights we gained on the potential that antifragile control could have in treatment design in clinical settings.

**Keywords:** antifragility; cancer; computational oncology; control theory

## 1. Introduction

Improving therapy outcomes in cancer care is still the crux of modern oncology. With cancers that become more robust and resistant to conventional schemes, therapy design undergoes a revolution. Tumors, where metronomic therapy paradigms were the gold standard, are now treated by sophisticated adaptive control algorithms and machine learning techniques capable to capture their growth and resistance patterns. The goal of such therapies is to induce controlled changes in the tumor environment through the timing and magnitude of the drug administration. Yet, such an intervention determines dramatic perturbations in the tumor's evolution and environment, but also in the structure of the surrounding normal cells vasculature, and, of course, the immune system response, as shown in Nia et al. [1]. Given the stochastic and complex nature of such processes, therapy schemes tend to fail in the face of the systemic variability and volatility of the effects they have upon the tumor. In other words, they are fragile. It is typically desired that such therapies would follow precise and fixed schedules designed by clinicians to control both the administration time and the dose magnitude to obtain a targeted response with minimal toxicity. However, the stochastic and patient-specific nature of the tumor-immune-drug dynamics impedes the development of suitable patient-centered interventions. There are already efforts combining mechanistic modeling and control algorithms targeted at enhancing the understanding and control of cancer dynamics by exploiting heterogeneous clinical data [2]. Such closed-loop systems can capture, as shown by Kurz et al. [3], and control,

as shown in Belfo et al. [4], the underlying dynamics of the physical interactions among the three main ingredients: the tumor, the immune system, and the chemotherapeutic drug and make treatments robust. In the recent seminal work of West et al. [5], antifragile therapy was introduced as a new framework to quantify the behavior of tumors in response to treatment variability, and how antifragile therapy can gain from such variability. The fundamental idea, promoted by this excellent work, is to identify and quantify regimes of fragility and antifragility through the curvature of the drug dose–response of the tumor. In other words, by analyzing the second-order behavior of the tumor dynamics in the presence of the drug, they were able to inform the optimality of a treatment schedule. Such treatment optimality included switching from even schedules in fragile tumor regions to uneven schedules in antifragile tumor regions. Despite the limited but promising results on lung cancer cell lines and 11 drugs, the framework has shed light on how to simultaneously capture the evolution of resistance patterns, the benefits of antifragile dosing, and their temporal evolution. Motivated by the great preliminary results of the antifragile therapy, we propose the development of adaptive antifragile therapy, which taps into treatment adjustments to control tumor growth and manage toxicity based on tumor responses that are clinically feasible and beneficial. While the approach of West et al. [5] allows for a straightforward prediction of optimal dose treatments for a large range of treatment schedules that amount to the same cumulative drug dose, our work will go beyond and enable a control-theoretic framework that can validate the link between the shape of the dose-response curve and the treatment schedule. More precisely, we seek to build a framework for treatment dosing and scheduling that can push the dynamics of the tumor-immune-drug network towards a patient-specific shape of dose response.

*The Need for Control in Cancer Therapy*

Cancer therapies follow national or international guidelines that are typically outlining the recommendations from the latest and most successful research studies on cumulative dosing, therapy staging, and therapy combinations. However, the therapy parameters, such as dose frequency and dose magnitude, are typically based on the prescribing physician's assessment of the specific tumor characteristics (i.e., BRCA1 and BRCA2 gene mutations in breast cancer for instance) and behavior (i.e., multistage carcinogenesis, from tumor genesis to metastasis).

The motivation to use control theory in therapy design stems from a couple of phenomena that describe the intricate dynamics of the tumor-immune-drug network, such as periodicity, resonance, and synchronization. For instance, cell growth is maximal when the periodicity of drug administration is a multiple of the characteristic periodicity of the cell population, as demonstrated in Kim et al. [6] and more visible in cell-cycle specific drugs, as described in McDonald et al. [7]. In this case, tumor growth is maximized when the periodicity of environmental perturbations acting upon the tumor is a multiple of the periodicity of the tumor population dynamics [8], and hence beneficial to exploit in drug administration. Nevertheless, such temporal phenomena can become more complex when the dynamics of the drug and the tumor-specific to a patient come into play. For instance, the work in Agur et al. [9] introduced a control theoretic approach based on resonance–antiresonance drug pulsing combined with vascularised tumor growth models for drug scheduling optimization and "drug dense paradigm" (in the Norton-Simon sense) that depends on the patient's cytokinetic and angiogenic parameters. Phenomena such as resonance can be used for timing the inter-dose interval to maximize the efficacy/toxicity ratio and minimize the toxicity/efficacy ratio (the dual problem).

Finally, when considering population-level interactions, covering both between-population competition (e.g., tumor–immune) and within-population cooperation (i.e., tumor–tumor), therapy design can exploit various phenomena which span across spatial scales. For instance, as cell population persistence depends on the local level of synchronization between the environmental and population processes (i.e., interaction among extracellular matrix and tumor) as shown in Agur et al. [10], the therapy could, in principle, induce resonance

through optimal drug dosing interval that maximizes host drug susceptible cells growth and minimizes malignant cells growth.

Optimizing chemotherapy strategies in order to maximize the efficacy/toxicity ratio can be done by exploiting the nonlinear tumor response dynamics and adaptive control of dosage under patient-specific tumor-immune-drug interactions. Yet, the complex dynamics describing patient-level tumor-immune-drug interactions are described by variability, volatility, and randomness. Control theory provides a systematic apparatus to tackle such a landscape, in order to ensure not just tolerating systemic variability but in fact gaining from it. This framework allows us to construct closed-loop systems, where the tumor (and its dynamics) is the system to control, given the external interactions with the immune system. The controller (also the core point of our contribution) computes the dose that drives the closed-loop system to the desired region in the reference/desired response curve. Figure 1 provides a general overview of our problem formulation and the context to support the mathematical apparatus, as well as a non-technical view suitable for physicians to understand the approach.

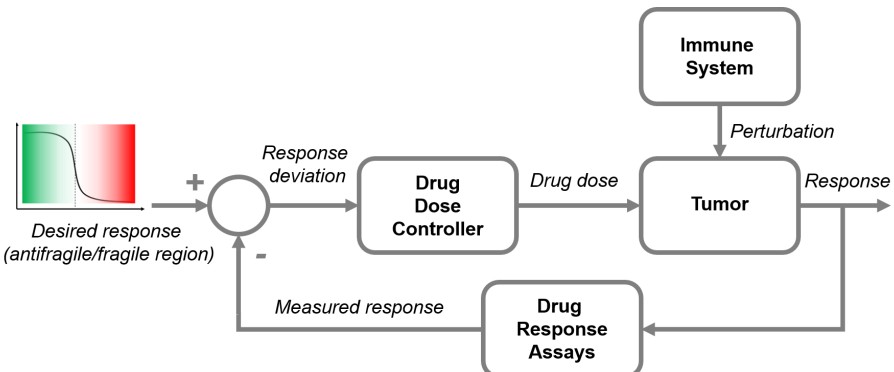

**Figure 1.** Therapy closed–loop control system: the main elements of our contribution in a control-theoretic feedback loop diagram. The purpose of the controller is to minimize the deviation from the desired response of the tumor by computing a suitable drug dose. The suitable dose can span from the fragile region ("red") to the antifragile region ("green") of the response curve under the effect of the controller output. The evolution of the tumor happens under the impact of the drug and also the immune system, which perturbs its evolution.

Such behavior is captured by the recently developed paradigm of antifragile therapy [5]. Instead of focusing on antifragility theory and heuristics to inform treatment scheduling or resistance management plans, we consider a systematic approach to building chemotherapy control systems that are antifragile.

This first computational oncology study is laying the basis for a novel framework for designing control systems that can operate in the presence of the randomness, variability, and volatility of the patient-specific tumor-immune-drug dynamics.

## 2. Materials and Methods

In this section, we first introduce the tumor-immune-drug interaction model based on the network design of dePillis et al., 2014 [11], which is further refined with the parametrization suggested in dePillis et al., 2003 [12] and dePillis et al., 2001 [13]. This model will be the subject of the control theoretic implementation of therapies based on various state-of-the-art approaches. Afterward, we formally introduce the antifragile control framework with its mathematical apparatus and controller synthesis details. Finally, we introduce the formalism and synthesis of typically used robust, optimal, and pulsed controllers for cancer therapy.

### 2.1. Tumor-Immune-Drug Network Model

To analyze and evaluate the antifragile control framework, we employ a network model that captures the tumor-immune-drug interactions from dePillis et al. [11]. The original model consists of three populations of cells (i.e., tumor population, immune cells population, and normal cell populations) which evolve through the mutual connections among themselves and under the presence/absence of the chemotherapeutic drug. The network model used in this study is depicted in Figure 2.

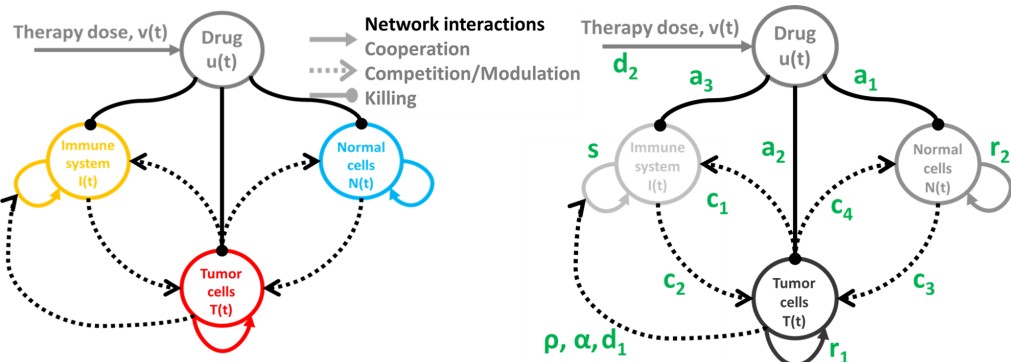

**Figure 2.** Network model of tumor-immune-drug interactions. Left panel: populations cells interactions; Right panel: differential equations parameters mapped on the network populations interactions. Each of the circles represents the respective cell population, the connecting lines three the types of interaction (i.e., cooperation—enhancement, competition—mutually induced decrease, and killing—extermination), and the impact of the administered drug.

The ordinary differential equations (ODE) describing the dynamics of the tumor-immune-drug network are introduced in Equations (1)–(4). The network interaction model captures competition (i.e., $c_1, c_2, c_3, c_4$ parameters) and modulation among tumor and immune and normal cells (i.e., $\rho, \alpha, d_1$ parameters, as well as the cooperation within each of the three populations (i.e., $s, r_1, r_2$ parameters), accounting for a basic form of self-excitation. The therapeutic drug acts upon all three cells populations in the network model by killing cells with a certain rate (i.e., $a_1, a_2, a_3$ parameters), as shown in the mapping in Figure 2—right panel. As visible in the network model, the presence of tumor cells stimulates the immune response, represented by the positive nonlinear growth term $\frac{\rho I(t) T(t)}{\alpha + T(t)}$ within the immune evolution in Equations (1)–(4).

$$\frac{dN(t)}{dt} = r_2 N(t)(1 - b_2 N(t)) - c_4 T(t) N(t) - a_1 (1 - e^{u(t)}) N(t) \tag{1}$$

$$\frac{dT(t)}{dt} = r_1 T(t)(1 - b_1 T(t)) - c_2 T(t) I(t) - c_3 T(t) N(t) - a_2 (1 - e^{u(t)}) T(t) \tag{2}$$

$$\frac{dI(t)}{dt} = s + \frac{\rho I(t) T(t)}{\alpha + T(t)} - c_1 I(t) T(t) - d_1 I(t) - a_3 (1 - e^{u(t)}) I(t) \tag{3}$$

$$\frac{du(t)}{dt} = v(t) - d_2 u(t) \tag{4}$$

For the readers interested to reproduce the analysis and experiments presented in the manuscript, we provide, in Table 1 in Section 3, the network model parametrization along with explanations on how each parameter impacts the dynamics of the overall model (as well as a link to the programs code). Recall that in our experiments the units of cells were rescaled. Hence, in our analysis one unit represents the carrying capacity of the normal cells in the region of the tumor. Although this depends on the type of tumor, we consider the reasonable scale on the order of $0.5 \times 10^{11}$ cells. This initial value is supported by physicians' remarks in dePillis et al., 2003 [12] and the model parametrization and analysis

in both works of Kuznetsov et al. [14] and dePillis et al., 2003 [12], which our model is based on.

Drug-Free Evolution

This section presents the drug-free evolution of the model to facilitate the analysis and the control design in the upcoming sections. In order to analyze the basic dynamics of the tumor-immune-drug network model, we consider the cumulative drug dose $u(t) = v(t) = 0$ when simulating the ODEs of the system. We start with a relatively large tumor burden $T(0) = 0.25$ and an analysis period of $t = 150$ days (i.e., a typical duration of a chemotherapy schema). The initial tumor size corresponds to a tumor with approximately $0.20 \times 10^{11}$ cells, or, in other words, a solid tumor of radius between 1.8 and 3.9 cm. To provide the reader with some reference, the clinical detection threshold for a solid tumor is typically around $10^7$ cells, this sets our chosen initial tumor volume of $0.20 \times 10^{11}$ above the clinical detection level. We also consider a patient with a weak immune system (but above the level of an immune-compromised patient) $I(0) = 0.1$. The immune threshold rate $\alpha$ is chosen inversely related to the immune response curve such that when the number of tumor cells $T$ is equal to $\alpha$ the immune response rate is at half of its maximum. In order to guide our control-theoretic design of therapy, we perform the null-space analysis of the drug-free dynamics. The evolution of each of the three interacting cell populations is depicted in Figure 3.

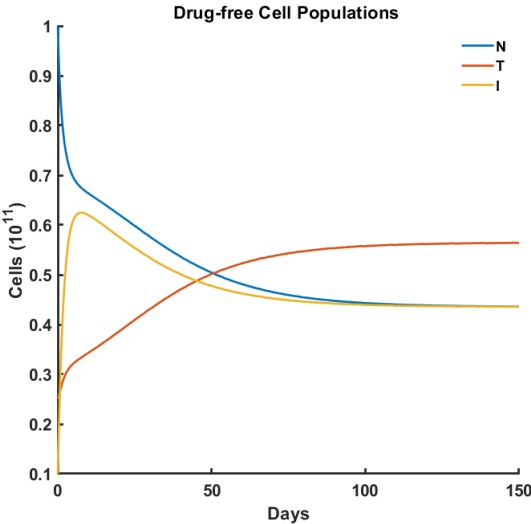

**Figure 3.** Drug-free evolution of the network model. The immune system (I) reacts by proliferating T-cells when the tumor (T) population (and the tumor itself) reaches a threshold triggering immune surveillance. The normal cells (N) production decreases as the tumor expands. In this case, without any drug administration, the tumor proliferates uncontrollably, escaping immune surveillance.

As we see in Figure 3, once the tumor has a considerable size (i.e., $T(0) = 0.20$), the immune system reacts with a very steep rise time in the first 10 days. The immune response is then modulated by the tumor proliferation and the decay in normal cell production which we see already after 50 days. This behavior is also visible in the configuration of equilibrium points in the dynamic landscape of the system (see Figure 4 for the null surfaces of each of the three cell types). Here, from the dynamical system analysis perspective, without any drug input, that is $u = 0$, the system must be either in the basin of a stable tumor-free equilibrium or in the basin of a stable equilibrium at which only a harmlessly small amount of tumor is present.

What we observe in the null-space analysis is that our network model can capture the multifaceted dynamics of the tumor–immune interactions. For instance, the drug-free setup allowed us to identify: the healthy tumor-free equilibrium $H$, the unstable interior

equilibrium $C_1$, the "unhealthy" interior stable equilibrium $C_2$, and two unstable "dead" equilibria where there are no normal cells $D_1$ and $D_2$, as shown in Figure 4. This description of null surfaces of the vector fields is useful in the characterization of the long-term behavior of all system's orbits. In our case, in order to consider the patient "cured", the system must be either in the basin of a stable "healthy" equilibrium or in the basin of a "unhealthy" stable equilibrium at which only a harmlessly small amount of tumor is present.

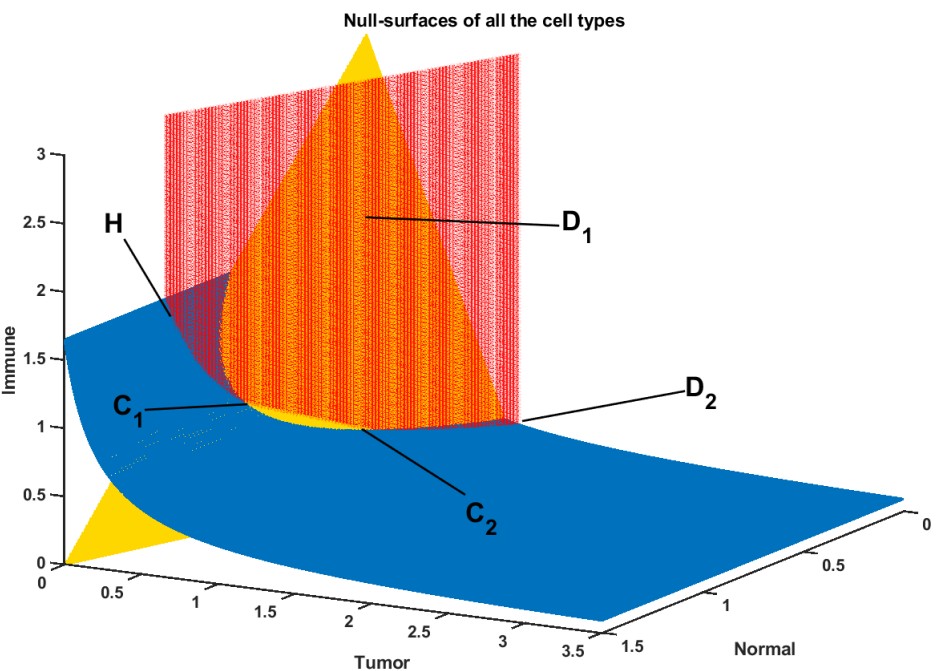

**Figure 4.** Dynamical null-space analysis of the drug-free model and the prerequisites for therapy control design. We are interested in having two attractors, one for "healthy" state (**H**) and one for "disease" (**C₂**) so that the control can push the dynamics to reduce tumor size under drug intake.

We further analyzed the model equilibria as a function of the immune cell influx without tumor $s$ and the immune response $\rho$—see Figure 5. This analysis is very important for the control design as it provides a sufficient characterization of the long-term behavior of all system orbits given the competitive, cooperative, and modulation interactions we consider in the model. Additionally, if the "healthy" tumor-free equilibrium is unstable then, according to the dynamics of the model in Equations (1)–(4), no amount of therapeutic drug will be able to completely eradicate the tumor.

Our findings are consistent with the experiments and analysis in dePillis et al. [12], especially when assessing the reasonable choice of the two stable attracting states, one of which is "healthy" (i.e., the **H** point in Figure 5) and another which is "diseased" (i.e., the **C₂** point in Figure 5). This is an intuitive design as a system with only a healthy attractor state would not need therapy and, conversely, a system with no healthy attractor state will never be cured, so no disease remission would be observed. We will see in the following sections how the basin of attraction of **H** will be affected by adding the drug to the network model and, later, the control theoretic calculation of the dosing $u(t)$.

We offer the reader the possibility to reproduce the analysis and reuse the models (i.e., tumor-immune-drug and control theoretic models) from the experiments performed in this study by downloading the code available on GitLab (Code available at https://gitlab.com/akii-microlab/antifragile-therapy-ctrl, accessed on 31 August 2022).

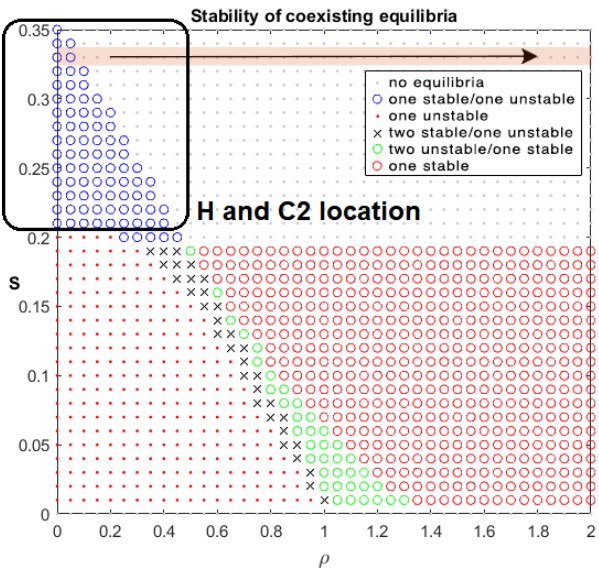

**Figure 5.** Model equilibria as a function of the immune cell influx without tumor *s* and the immune response *ρ*. The control design will revolve around the marked region in the landscape of coexisting equilibria with two stable attractors, one for "healthy" state (**H**) and one for "disease" (**C₂**).

### 2.2. Antifragile Control

In this section, we describe the mathematical apparatus of antifragile control, going from the theory and principles to the control synthesis.

#### 2.2.1. Preliminaries

In this subsection, we offer a short primer on the principles, theory, and design of antifragile control systems, with a focus on the cancer therapy problem. As coined in the book of Taleb [15], antifragility is a property of a system to gain from uncertainty, randomness, and volatility, opposite to what fragility would incur. An antifragile system's response to external perturbations is beyond robust, such that small stressors can strengthen the future response of the system by adding a strong anticipation component.

When looking at control systems, inducing such behavior to a feedback control loop accounts for a novel design and synthesis approach where: (1) redundant overcompensation can bring the system into an overshooting mode that builds extra-capacity and strength in anticipation; (2) structure-variability can induce stressors and carry inherent information that emerges only under volatility and randomness of the system dynamics altered by the application of a high-frequency switching control signal; and (3) high-frequency control activity opposite to a "tight control" which only eliminates the beneficial effect of noise and volatility of the closed-loop dynamics. We hereby guide the reader on how the above concepts related to the therapy control problem and their practical instantiation.

Antifragile control is designed to capture and exploit the relations between the non-linearity of drug dose–response and the properties of the outcomes (i.e., tumor kill ratio, toxicity ratio, and their average and variations) under complex tumor-immune-drug dynamics. But how can one quantify antifragility? In his work, Taleb [16] provided a clear mathematical and, rather geometrical, description: "antifragility" is described as the mathematical property of a local convex response and its generalization, whereas the designation "fragility" is its opposite, the locally concave response.

This perspective was further refined by West et al. in [5] where the drug dose–response shape was used to denote a fragile or antifragile therapy scheme, respectively. More precisely, considering a typical drug dose–response, namely the sigmoidal-shaped Hill function, described in Goutelle et al. [17], indicating the percent of cells that survive a given dose, the tumor's response to treatment is antifragile if the curvature is negative, otherwise

is fragile. In short, one can consider that antifragility is a second order effect, whereas, for instance the average is the first order effect.

Provided this intuition of how antifragile control can be designed, we elaborate, in the following subsections, the formalism and the controller synthesis for therapy control.

### 2.2.2. Formalism

As shown in the previous section, the shape of the drug dose–response is a useful heuristic to detect antifragility and, implicitly, inform tumor evolution and optimal dosing plans within a therapy schedule. Yet, assessing the drug dose–response in complex tumor-immune-drug models, such as the one we consider in Equations (1)–(4), is highly dependent on the cell populations dynamics and interactions. For instance, let's consider the tumor cells. In solid cancers, they form heterogeneous populations with varying drug sensitivities that depend on multiple factors such as the cell cycle stage, as shown in Gaffney et al. [18], the presence of geno- or phenotype features, as described in Fedorinov et al. [19], and the environmentally mediated drug resistance, modeled in Paraiso et al. [20].

Under the antifragile control of drug dose, the tumor-immune-drug model is a dynamical system whose state space is not a vector space but rather a curved state manifold (i.e., the drug dose–response surface), more precisely, a place and a dynamic to push the system towards, as shown in the theoretical formulation of Bejenaru et al. [21]. To get an intuition, we revisit the attractor states analysis in Figure 4, where the intersecting planes of the tumor, immune, and normal cell populations evolution describe the geometrical properties of the model dynamics. We aim at designing a controller that forces the dynamics to evolve on a prescribed state manifold with certain geometrical properties. Control systems on state manifolds is a relatively new research branch in non-linear control theory, with a very well formalized mathematical apparatus cast in the Riemannian geometry, introduced by the work in Lee et al. and Bloch et al. [22,23].

For our therapy control problem, we want to push the tumor-immune-drug system to the desired dynamics of the drug dose–response surface, turning this into a tracking control problem. Recall that we want to have two attractor states lying on manifolds whose curvature (i.e., second derivative) is informative on the joint populations' dynamics. More precisely, the curvature of the imposed dynamics surface can be assessed on Riemannian manifolds by using, for instance, the Penot generalized directional derivative, and the Clarke generalized gradient, proposed by Zou et al. [24]. This enables us to formulate the main pillars of antifragile control: redundant overcompensation, structure variability, and bounded high-frequency control activity, in terms of Riemannian geometric control, as suggested by Bullo [25]. For instance, having the advantage that translation in Euclidian space is a canonical transport with acceleration and velocity in Riemannian space, we can apply the exponential map to perform updates along the shortest path in the relevant direction in unit time while remaining on the desired drug dose–response manifold region, as suggested by Becigneul et al. [26].

#### Dynamical Systems on Manifolds

In the following, we recall notions from manifold calculus and illustrate how these tools prove useful in describing the system-theoretic design of an antifragile controller. We describe redundant overcompensation, structure variability, and bounded high-frequency control activity using manifold calculus which: (1) carries the advantage that is coordinate-free; (2) relies on the embedding of a manifold into a larger dynamical space; (3) supports simpler definitions of control of non-linear systems whose states belong to curved manifolds, as shown by Fiori et al. [27].

Let $Q$ denote a Riemannian manifold. At each point $x \in Q$ is possible to define the tangent space $T_x Q$, i.e., a real vector space containing all possible directions tangentially passing through $x$. The disjoint union of the tangent spaces of $Q$ forms the tangent bundle, formally defined as $TQ := \{(x,v) | x \in Q, v \in T_x Q\}$. The Riemannian manifold $Q$ is equipped with a bilinear, positive-definite inner product operator $\langle ., . \rangle_x : T_x Q \times T_x Q \to \mathbb{R}$.

This also defines a local norm $\|x\|_x := \sqrt{\langle v, v \rangle_x}, v \in T_x Q$. The Riemannian gradient of a function $f : \mathbb{R} \to Q$ at a point $x \in Q$ is $\frac{df}{dx}$. The exponential map $exp$ of a manifold $Q$ is a function $exp : TQ \to Q$ which takes a point-velocity pair $(x, v) \in TQ$ and maps it to a point on the manifold $Q$, namely $exp_x(v) \in Q$. The inverse, the manifold logarithm $log$ is defined only locally and returns a vector $v = log_x(y) \in T_x Q$ such that $exp_x(v) = y$, given $x, y \in Q$. In order to compute a distance $d(x, y)$ on the manifold $Q$, we define a geodesic arc as the manifold-theoretic counterpart of a straight line on a curved manifold. On a Riemannian manifold, the distance between two points may be calculated as $d(x, y) = |log_x(y)|_x$ and implicitly the gradient of the squared distance as $\frac{d^2(x,y)}{dx} = -2log_x(y)$, given, of course, that the logarithm is defined. Finally, we assume the manifold $Q$ to be equipped with a metric connection (which makes it easier to compute the covariant derivative involving simply an inner product). The covariant derivation operator $\nabla_v w$ of a vector field $w_x \in T_x Q, x \in Q$ in the direction of a vector $v \in TxQ$ is closely related to the transport map $T_{x,y}$ as one can express the derivation in terms of parallel transport as

$$\nabla_v w \approx \frac{T_{(\gamma(h) \to x)}(w_{\gamma(h)}) - w_x}{h} \tag{5}$$

$$\gamma(0) = x \in Q, \dot{\gamma}(0) = v \in T_x Q \tag{6}$$

$$h \ll 1 \tag{7}$$

In this study, the purpose of the antifragile controller is to drive the state $q(t) = [I(t), T(t), N(t), u(t)]^T$ of the tumor-immune-drug system toward a reference point on the desired/reference configuration $r(t)$—capturing the dynamics of the system in the antifragile region of the drug dose–response surface (see West et al. [5]). The first step in the controller design is the definition of a control error function, quantifying how far is the system state from the reference.

Control Error Function

The control error function, $\varphi : Q \to \mathbb{R}$, describes the distance between the reference state (i.e., the tumor-immune-drug system dynamics in the antifragile region on the desired drug dose–response shape) denoted by $r$ and the actual configuration/state $q$ on a state manifold $Q$. In our case, the reference dynamics is determined by the position of the tumor-immune-drug system trajectory on the drug dose–response curve that describes the survival rate $S$ vs. drug dose $u$. Here, a lower survival is better as it accounts for more tumor kills and reaches the antifragile control region. In our case, we consider $\varphi$ a uniformly quadratic function with constant $L$. This implies that, for all $\epsilon > 0$, there exist $b_1 \geq b_2 > 0$ such that

$$b_2 \left\| \frac{d\varphi(q,r)}{dq} \right\|^2 \leq \varphi(q,r) \leq b_1 \left\| \frac{d\varphi(q,r)}{dq} \right\|^2 \tag{8}$$

In our case, we consider the configuration $q(t) = [I(t), T(t), N(t), u(t)]^T$ and the reference $r(t) = [I_S(t), T_S(t), N_s(t), u_S(t)]^T$. The control error $\varphi$ is the most important component in ensuring the redundant overcompensation through its temporal evolution $\varphi(t)$ and its velocity $\dot{\varphi}(t)$.

Parallel Transport Map

Given the configuration $q(t)$ and the reference $r(t)$, as defined above, a linear map $P_{(q,r)} : T_r Q \to T_q Q$ is a parallel transport map if it is compatible with the error function $\varphi$, that is

$$\frac{d\varphi(q,r)}{dr} = -P_{(q,r)} \frac{d\varphi(q,r)}{dq} \tag{9}$$

Intuitively, the parallel transport map describes the actual dynamics transformation that pushes the system towards the desired manifold along a geodesic. Now, given the transport map $P$, the velocities (i.e., first derivatives) of both the configuration $q$, marked as $\dot{q}$ and reference $r$, marked as $\dot{r}$, can be compared to build a velocity error, i.e.,

$$\dot{e} = \dot{q} - P_{(q,r)}\dot{r} \in T_q Q \tag{10}$$

Given the previous notations, we can now compute the time derivative of the error function $\varphi$ such as

$$\frac{d}{dt}\varphi(q(t), r(t)) = \frac{d\varphi(q(t), r(t))}{dq(t)}\dot{e}(t) \tag{11}$$

These geometrical objects serve as tools to introduce the system-theoretic synthesis of the antifragile controller in the following section.

### 2.2.3. Control Synthesis

In the current section, we formally introduce the three key components for antifragile control synthesis, namely redundant overcompensation, structure variability, and bounded high-frequency control activity.

### Redundant Overcompensation

We start the synthesis of the redundant overcompensation component which will induce the anticipation capabilities of the antifragile controller. From a control theoretic point of view, this accounts for a combined Proportional Derivative (PD) control action. Let's start with the more general form of our tumor-immune-drug model as a dynamical system in the Riemannian manifolds framework such as

$$\dot{q}(t) = v(t), t \geq 0 \tag{12}$$
$$\nabla_{\dot{q}}\dot{q} = \mathbb{S}(t, q(t), v(t)) + F(q(t)) \tag{13}$$

where $q$ is the state, $\dot{q}$ is the velocity of the state, $\mathbb{S}$ is the time varying-state-transition map $\mathbb{S} : \mathbb{R} \times TQ \to TQ$, $\nabla$ is the Riemannian covariant derivative, and $F$ the control law.

As in the "traditional" PD control design, we need the error function between the state $q$ (i.e., configuration) and the reference $r$, as well as the anticipation component. We consider the previously introduced error function $\varphi$ on the manifold $Q$. The antifragile proportional gain $K_\varphi$ is a smooth self-adjoint positive tensor filed on the manifold $Q$ and $K_q : T_q Q \to T_q^* Q$ is the antifragile derivative gain. Putting all together, the antifragile PD control law for therapy design is computed as

$$F(q) = K_\varphi \varphi(q) + K_q \dot{\varphi}(q) \tag{14}$$

where $q(t) = [I(t), T(t), N(t), u(t)]^T$. This law is consistent with similar approaches for simple PD control on manifolds, described in Fiori et al. [28]. As a trademark of antifragile control, the first term in the Equation (14), $K_q \dot{\varphi}(q)$, seeks to anticipate (not to predict) a higher level of error than the previous maximum through a redundant overcompensation that builds extra-capacity through the choice of the transport map that determines the term $K_q \dot{\varphi}(q)$. This component is responsible to drive the system closer to the desired dynamics, so we now evaluate the (Lyapunov) stability of the closed-loop given the control law $F(q)$. We start by selecting a Lyapunov candidate function

$$V(\varphi, q) = \varphi + \frac{1}{2}\|\dot{q}\|^2 \tag{15}$$

that is positive definite in $q$ and $\dot{q}$. Then, the derivative of $V$ is calculated as

$$\frac{dV(\varphi, q)}{dt} = \frac{d}{dt}(\varphi + \frac{1}{2}\|\dot{q}\|^2) \tag{16}$$

$$= \nabla_{\dot{q}}\varphi + \frac{1}{2}\nabla_{\dot{q}}\|\dot{q}\|^2 \tag{17}$$

$$= \langle d\varphi, \dot{q}\rangle + \langle \nabla_{\dot{q}}\dot{q}, \dot{q}\rangle \tag{18}$$

$$= \langle d\varphi, \dot{q}\rangle + \langle -K_\varphi\varphi(q) - K_q\dot{\varphi}(q), \dot{q}\rangle \tag{19}$$

$$= -\langle K_q\dot{\varphi}(q), \dot{q}\rangle \tag{20}$$

Then, given that $V(\varphi, q)$ is positive definite and $\frac{dV(\varphi, q)}{dt}$ is negative semi-definite, the closed-loop system is stable in the Lyapunov sense. For the asymptotic stability one can use the LaSalle invariance principle.

Stucture Variability

While there are many advanced approaches, such as adaptation based on response identification and state observation or absolute stability techniques, the most straightforward way to deal with uncertainty is to keep certain limitations by "brute force". However, any carefully maintained equality eliminates one "uncertainty dimension". The theory of variable structure control (VSC) developed around this principle, formalized in Guo et al. [29] and opened up a wide new area of development known as sliding mode control (SMC), that is characterized by a discontinuous control action which changes structure upon reaching a set of predetermined switching surfaces, such as in the work of Colli et al., and Ouyang et al. [30,31]. The main advantages of using SMC in our therapy control design are listed below:

- the motion equation of the sliding mode, as nicely framed in Slotine et al. [32], can be designed linear and homogeneous, despite that the tumor-immune-drug model is governed by nonlinear equations,
- the sliding surface does not depend on the process dynamics, but it is determined by parameters selected by the designer, as suggested in deCarlo et al. [33], i.e., desired trajectory of the system in the antifragile region of the drug dose–response curve (e.g., Hill function),
- once the sliding motion occurs (i.e., the system dynamics are on the surface), the system has invariant properties which make the motion independent of certain system parameter variations, uncertainty, and disturbances, as described in the famous work of Utkin [34]. Hence, the system performance can be completely determined by the dynamics of the sliding manifold.

The first design element in the variable structure SMC is the choice of the sliding surface. Starting from the control error function $\varphi$, we define $S$ as a time-varying surface in the state space/configuration space $q$, where

$$S(q, t) = \left(\frac{d}{dt} + \lambda\right)^{n-1} \varphi, \ \lambda > 0. \tag{21}$$

For instance, if $n = 2$ then $S(q, t) = \dot{\varphi} + \lambda\varphi$, if $n = 3$ then $S(q, t) = \ddot{\varphi} + 2\lambda\dot{\varphi} + \lambda^2\varphi$, and so on. As we see in Figure 6b, the sliding surface is a curve in $(q, \dot{q})$ space of slope $\lambda$ and containing the time-varying reference configuration $r(t)$. Recall that, for the "redundant overcompensation", the system follows the time-varying state transition $\mathbb{S}$. In SMC, the controller needs to "force" the system trajectories to "move" while still pointing towards the surface as depicted in Figure 6a,b and theoretically proven in the work of

Slotine et al. [32]. In other words, the curvature needs to decrease along the system trajectories such that

$$\frac{1}{2}\frac{d^2 S}{dt^2} \leq -\eta|S|, \; \eta > 0. \tag{22}$$

The sliding condition in Equation (22) makes the surface $S$ an invariant set, which is both a place in the $(q, \dot{q})$ space as well as a dynamics that the system will follow once reaching the surface, i.e., when $\mathbb{S} = S$ (see Equation (13)).

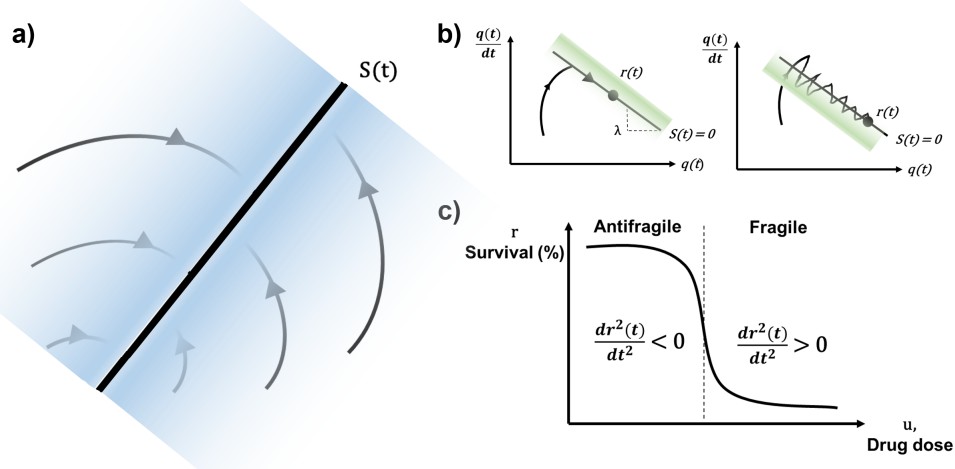

**Figure 6.** Variable structure control using SMC. (**a**) the sliding condition, from initial states, the system is pushed towards the surface $S$; (**b**) the dynamics on the sliding surface; (**c**) drug dose–response curve with fragile and antifragile regimes depending on curvature (adapted with permission from West et al. [5]).

Particularized to antifragile therapy control, the desired configuration can be given by a vector $r(t)$ which will be "pushed", given the control law $F$ to the convex (i.e., antifragile) region of the drug dose–response depicted in Figure 6c and mathematically described as

$$r = r_{min} + \frac{r_{max} - r_{min}}{1 + \left(\frac{u}{\mu}\right)^{-n}}. \tag{23}$$

For the definition of the survival in Equation (23), we consider the Hill function with $r_{min}$ minimal survival and $r_{max}$ maximal survival, $\mu$ is the inflection point of $r$ beyond which increases of a drug have less impact on survival, and $n$ the Hill exponent. Note that the Hill function is a commonly employed mathematical model used to parameterize dose-response assays, as shown by Meyer et al. [35], although other functions might be used. In our experiments, we consider the parametrization in West et al. [5], where: $r_{min} = 20$, $r_{max} = 100$, $n = 10$, $\mu = 10$.

The reference state $r(t)$ (i.e., the configuration) is computed such that the dynamics of the tumor-immune-drug dynamics network model are driven to the antifragile region of the survival curve in Figure 7a using an antifragile control law synthesized through the combined effect of the anticipation $K_q \dot{\varphi}(q)$ and the variable structure component $\beta \operatorname{sgn}(S)$ of the controller, which ensure the curvature decrease along the system state trajectory (see green region in Figure 7b).

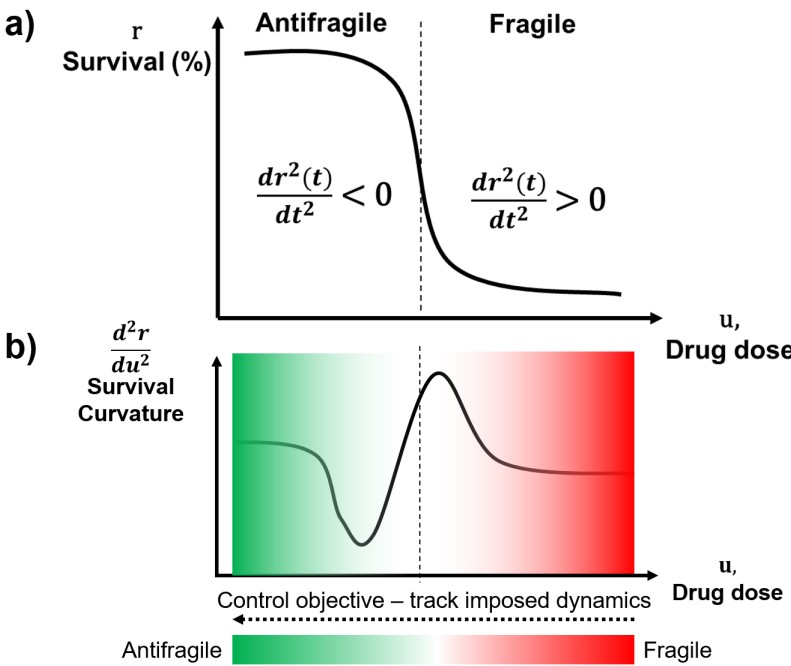

**Figure 7.** Antifragile principle. (**a**) The drug dose–response basis for the imposed dynamics of the network. The survival function assumes that the system needs to provide a drug dose such that the value of *r* decreases; (**b**) The derivative of the survival function. This demonstrates that the inflection point describes the point where the dosage (control law) can drive the tumor-immune-drug system in the antifragile region (adapted with permission from West et al. [5]).

Putting all together, and hence combining the redundant overcompensation of the PD component in the antifragile design with the variable structure of the SMC, we have the following expression of the control law,

$$F(q) = K_\varphi \varphi(q) + K_q \dot{\varphi}(q) + \beta \, \mathrm{sgn}(S), \tag{24}$$

where $\beta$ is the SMC control gain and $\mathrm{sgn}(S)$ is the sign function applied to the chosen sliding surface $S$, parametrized such that the condition in Equation (22) is fulfilled.

Bounded High-Frequency Activity

The final step in our controller design is the computation of the control law $F$ that verifies the sliding condition in Equation (22) and is discontinuous, in order to account for uncertainty, volatility, and disturbances. Although the choice of the control law is done with a trade-off between control bandwidth and tracking precision in mind (see the last term in Equation (24)), we need to ensure the presence of the low-amplitude high-frequency "stressors" that will "shake" the system path towards the antifragile region (see Figure 6b,c. The standard SMC control law will cause the controlled system chattering due to the switching action of the control law, as shown by Zou et al. [24], which is typically a function of the form $\beta \, \mathrm{sgn}(S)$, where the SMC control gain $\beta = \beta(q, \dot{q})$ chosen such that Equation (22) is satisfied and

$$\mathrm{sgn}(S) = \begin{cases} 1 & \text{if } S > 0 \\ 0 & \text{if } S \leq 0 \end{cases}. \tag{25}$$

To reduce a high chattering value, which might excite high-frequency structural modes, neglected time-delays, or other modeling uncertainties, a saturation function can be chosen to replace the sgn function such as

$$\text{sat}(S) = \begin{cases} \text{sgn}(S) & \text{if } |S| \geq \Phi \\ \frac{S}{\Phi} & \text{if } |S| < \Phi \end{cases}. \tag{26}$$

This ensures that the control error $\varphi$ is maintained within a guaranteed precision $\epsilon$ calculated as $\epsilon = \frac{\Phi}{\lambda}$. We build the sliding surface $S$ as a linear combination of the control error $\varphi$ and its velocity $\dot{\varphi}$ such as $S = \dot{\varphi} + \lambda\varphi$. Then, the control law in Equation (24) becomes

$$F(q) = K_\varphi \varphi(q) + K_q \dot{\varphi}(q) + \beta \, \text{sat}(\dot{\varphi}(q) + \lambda\varphi(q)). \tag{27}$$

Finally, given the antifragile controller we synthesized, we will show, throughout our experiments in the next section, that depending on the curvature of the drug-response reference dynamics on the manifold we impose, we can identify regions of "fragile" response in which the synthesized control law has low fluctuations, and regions of "antifragile" response in which the control law has large dose fluctuations, consistent with the work of West et al. [5]. But in order to achieve this behavior the choice of $\varphi$ and $T$ is very important, as postulated by Maithripala et al. [36]. For instance, $\varphi$ determines how fast $q(t)$ reaches $r(t)$ in the topology induced by $\varphi$, whereas $T$ determines how "simple" is the control law, as proposed in the studies of Zou et al. and Zhang et al. [24,37]. This will subsequently impact the computation of the sliding surface $S$ and the overall characteristics of the control law $F$, which in our model accounts for the drug dose $v$.

Note that the survival dose curve parametrizes the reference configuration $r(t)$ in the control problem (see Figure 7a) such that the antifragile region in Figure 7b is attained by minimizing the error function $\varphi$ given the actual state of the tumor-immune-drug network model in Equations (1)–(4). In the antifragile control design the anticipation control part (i.e., PD controller) mainly contributes to the normal stabilization of the tumor-immune-drug system, forcing the tracking errors in the boundary layer of the sliding surface through a redundant overcompensation and high control activity (see Figure 8). After entering the boundary layer of the surface (see Figure 6b), the tracking performance is dominantly controlled by the SMC control part, described by $\beta \, \text{sat}(\dot{\varphi}(q) + \lambda\varphi(q))$, which pushes the trajectory on the sliding surface (see Figure 6a,b).

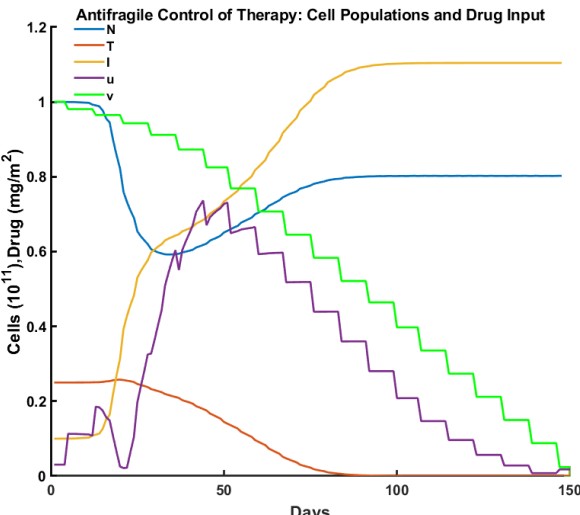

**Figure 8.** Antifragile control therapy. Individual cell populations evolution under the computation of the antifragile control law of drug dose $v$. The antifragile drug dose timing and duration are computed based on the control law in Equation (27).

*2.3. State-of-the-Art Control Algorithms in Cancer Therapy*

Control-theoretic therapy design is used to describe treatment protocols that have the potential to be more efficient (i.e., maximize tumor kill, minimize patient toxicity) than standard static periodic protocols now in use, as suggested in the review of Lecca [38]. As discussed in Section 1, static therapy plans cannot cope with the highly nonlinear phenomena emerging in tumor-immune-drug network models, such as periodic oscillations in tumor and normal cells generation, presented in the work of both Kim et al., and Hu et al. [6,8], the resonance–antiresonance in drug response and toxicity, from the great work of Schattler et al., Agur et al., and Li et al. [2,9,39], or the synchronization in tumor–environment cells interactions under drug pulses, modeled in the work of Ren et al., and Agur et al. [10,40]. In order to handle such phenomena intrinsic in the tumor-immune-drug network, many control theoretic approaches were developed since the '80s, with notable work of Swan et al., and dePillis et al. [12,41] and up to recent work of Carrere et al., and Irurzun-Arana et al. [42,43].

Although employing different control laws, the core objective of all the approaches is to move the system into the basin of attraction of a "healthy" stable equilibrium state under the impact of drug dosing and timing, demonstrated in the work of Uthamacumaran et al. [44] which is also suitable for surgical intervention, as shown in Axenie et al. [45]. In a more general view, this accounts for a state motion towards a reference manifold under the control law action, which then turns the manifold into an invariant set (i.e., the system is described by the manifold's equation).

2.3.1. Optimal Control

In the optimal control synthesis the goal is to determine the therapy dose function $v(t)$ in Equation (4), representing the chemotherapy administration schedule, as introduced in the works of dePillis et al. and Wang et al. [12,13,46]. This is computed such that the kill rate of the tumor cell population is as high as possible, with the constraint that the killing rate of normal cells is minimized. Although for our experiments we consider the simplified approach in the study of dePillis et al. [12], the general optimal control problem can be stated as follows.

The goal is to find the control variable, in our case the therapy dose $v(t)$ in Equation (4), and the (possibly free) final time $t_f$, corresponding to the end of therapy (e.g., after 150 days), that solves the following optimization problem

$$
\begin{aligned}
\underset{v,t_f}{\text{minimize}} \quad & J(v,t_f) = \varphi(x(t_f),t_f) \\
\text{subject to} \quad & \dot{x}(t) = f(x(t),v(t),t), \; t_0 \leq t \leq t_f, \\
& g(x(t)) \geq 0.
\end{aligned}
\tag{28}
$$

In Problem (28), objective function $J = \varphi(x(t_f),t_f)$ is the tumor burden of the patient given the changes in the interacting cell populations $N,T,I$ described by $x(t)$ and their evolution $f(x(t),v(t),t)$ under the drug administration. The state constraint $g(x(t))$ is used to maximize tumor kill and keep normal cells above a threshold. This formulation keeps the tumor cell population $T$ lower at $t_f$ at the price of large oscillations.

We update the problem such that we can weight each component of the objective function to avoid oscillations in the network model of Equations (1)–(4). We then rewrite the objective function $J$ as a weighted combination of tumor burden at therapy end $T(t_f)$, the summed tumor burden over the treatment period $\int_0^{t_f} T(t)dt$, the maximum tumor burden over the treatment course $\max_{t \in t_o,t_f} T(t)$, and, of course, the drug dose concentration $u(t)$ during the therapy duration. The updated optimal control objective function $J$ is given as:

$$
J(v,t_f) = w_1 T(t_f) + w_2 \int_0^{t_f} T(t)dt + w_3 \max_{t \in t_o,t_f} T(t) + w_4 u(t),
\tag{29}
$$

where $w_i$ are weighting constants, chosen as $w_1 = 1500, w_2 = 150, w_3 = 1000, w_4 = 40$. To evaluate the effect of the optimal control law has upon the tumor-immune-drug model we plot the evolution of the system in Figure 9. The comparative evaluation and discussion will be done in Sections 3 and 4.

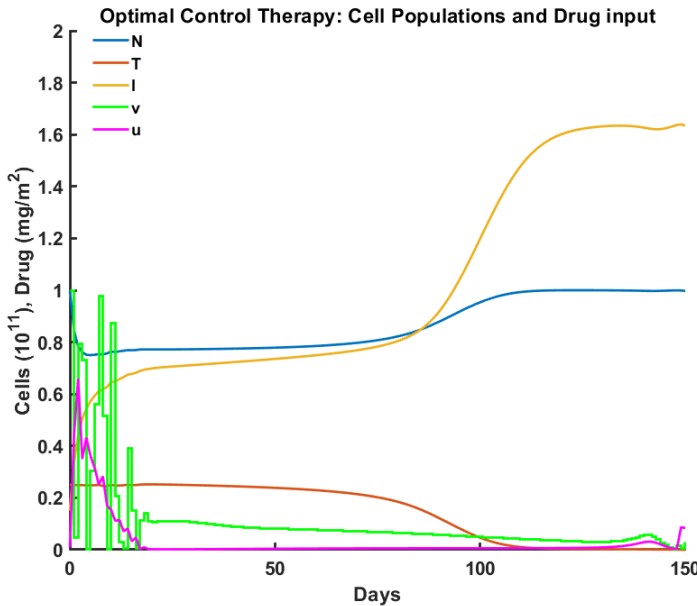

**Figure 9.** Optimal control therapy. Individual cell populations evolve under optimal computation of the drug dose $v$. The drug dosing amplitude, timing, and duration is computed as a constrained optimization problem that slows down and, eventually, disrupts tumor growth. When the tumor (T) decays under the detectable size ($\approx$80 days), the normal cells population (N) restarts the normal proliferation pattern, and immune reaction (I) is enhanced to support the complete tumor extinction.

2.3.2. Robust Control

Another line of research, which goes beyond the optimal control landscape, is the variable structure control, such as bang-bang control proposed by Ledzewicz et al. [47]. This approach allows for seamless compensation of un-modeled dynamics in the tumor-immune-drug model and parametric uncertainties alike, as suggested by Ledzewicz et al. [48,49]. Such models capture the dynamics of the phase field model of tumor growth, as suggested by Colli et al., and Guo et al. [29,30] and the regularities of the cell population states to control the phase of the drug concentration. In our study, we considered the model of dePillis [13] with parametrization considerations from Ledzewicz et al. [47].

In our therapy design problem, the control signal $v$ is restricted to be between a lower and an upper bound of dosage $0 \leq v(t) \leq v_{max}$, as suggested by oncology guidelines. Additionally, in a special case of the optimal control in Problem 28, $v$ switches from one extreme to the other (i.e., is strictly never in between the bounds). This is referred to as a bang-bang therapy solution, explored in both Ledzewicz et al. and dePillis et al. [13,47]. In order to formulate the bang-bang control synthesis, we derive the control Hamiltonian (Note that, for the therapy control design we use the control Hamiltonian that describes the conditions for optimizing some scalar function, basically the Lagrangian with respect to a control variable, and not the dynamics of the system itself.) of the optimal control problem (see Equations (30)) in order to identify the switching function of the system (i.e., ensuring that that the drug entering the patient at time $t$ is bounded).

$$H = \lambda_1 \left(\frac{dI}{dt}\right) + \lambda_2 \left(\frac{dT}{dt}\right) + \lambda_3 \left(\frac{dN}{dt}\right) + \lambda_4 \left(\frac{du}{dt}\right) + \eta, \tag{30}$$

where the functions $\lambda_i$ satisfy the co-state variables equations

$$\frac{d\lambda_1}{dt} = -\lambda_1\left(\frac{\rho T}{\alpha + T} - c_1 T - d_1 - a_1(1 - e^{-u})\right) + \lambda_2 c_2 T,$$

$$\frac{d\lambda_2}{dt} = -\lambda_1\left(\frac{\rho \alpha I}{(\alpha + T)^2} - c_1 I\right) - \lambda_2(r_1 - 2r_1 b_1 T - c_2 I - c_3 N - a_2(1 - e^{-u})) + \lambda_3 c_4 N,$$

$$\frac{d\lambda_3}{dt} = \lambda_2 c_3 N - \lambda_3(r_2 - 2r_2 N^2 - c_4 T - a_3(1 - e^{-u})) - \eta(t),$$

$$\frac{d\lambda_4}{dt} = -e^{-u}(a_1\lambda_1 I + a_2\lambda_2 T + a_3\lambda_3 N).$$

$$(31)$$

The $\eta$ is chosen to ensure that the normal cells $N$ are above 75% of the tumor-free normalized carrying capacity for this experiment such that

$$\eta(t) = \begin{cases} 1 & \text{if } N \leq 0.75 \\ 0 & \text{otherwise} \end{cases}. \tag{32}$$

We can now rewrite the robust control equation for the drug dosing as

$$\frac{\partial H}{\partial v} = \lambda_4, \tag{33}$$

which is independent of the control variable $v$. If we assume that the amount of drug entering the patient at time $t$ is bounded above and satisfies $0 \leq v(t) \leq v_{max}$, then the output of the bang-bang control is given by

$$v(t) = \begin{cases} 0 & \text{if } \lambda_4 > 0 \\ v_{max} & \text{if } \lambda_4 < 0 \\ \text{singular} & \text{if } \lambda_4 = 0 \end{cases}. \tag{34}$$

As we can see in Equation (34), the co-state variable $\lambda_4$ is the switching function for the tumor-immune-drug network model we consider. In this case, the drug should be injected at the maximum rate when $\lambda_4$ is negative and should be ceased when $\lambda_4$ is positive. To evaluate the effect the robust control law has upon the tumor-immune-drug model, we plot the evolution of the system in Figure 10. The comparative evaluation with the other state-of-the-art methods and the antifragile control will be done in Sections 3 and 4.

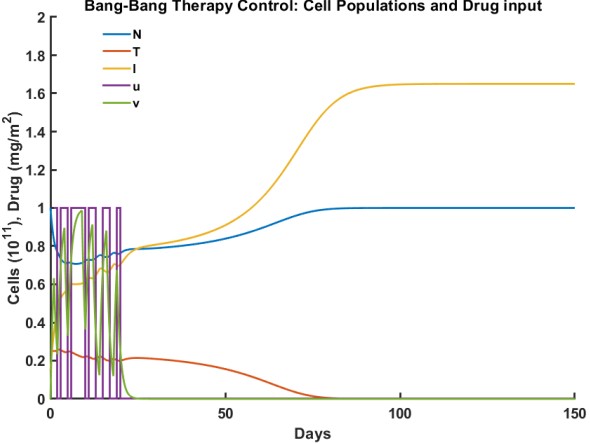

**Figure 10.** Robust control therapy. Individual cell populations evolution under the computation of the robust control law of drug dose $v$ based on Equations (31). The bang-bang drug dose timing and duration are computed such that the tumor population (T) decreases rapidly after treatment cessation ($\approx 50$ days). This decrease in tumor size is also determined by the exponentially increasing immune response (I) and marks also the normal cell (N) proliferation/death (i.e., through toxicity) pattern.

### 2.3.3. Pulsed Control

Pulsed control therapy assumes that the administration of the drug into the patient body is modeled by a train of Dirac impulses. This approach is the traditional approach to chemotherapy and was approached by multiple studies, such as the ones from Panetta et al., Ren et al., and Belfo et al. [4,40,50]. In order to parametrize the control system one needs to specify: (1) the total number of treatment sessions $N$ (basically the total number of impulses); (2) the time interval between treatment sessions $Tn$ (inter-impulse time interval); and (3) the drug dose administered in each session $An$ (accounting for the amplitudes of the impulses).

Despite its simplicity, this control method is not applied in isolation, as suggested in Belfo et al. [4]. The control variable $v$, represented by a train of Dirac impulses, is typically computed in the optimal control framework, as modelled in Ren et al. [40]. Additionally, the control design problem may be reduced to a finite-dimensional optimization problem that can be handled by a suitable solution, since the control variable linearly influences the model, and we follow an a-priori known number of impulses. To evaluate the effect the pulsed control law has upon the tumor-immune-drug model, we plot the evolution of the system in Figure 11. The comparative evaluation with the other state-of-the-art methods and the antifragile control will be done in Section 4. Finally, the fundamental optimal impulsive control sequence is utilized in a receding horizon strategy to produce a feedback control law. This strategy entails applying to the patient only the first control action of the sequence and repeating the entire dynamic optimization process beginning with the patient's state at the subsequent discrete time instant as shown in Belfo et al. [4].

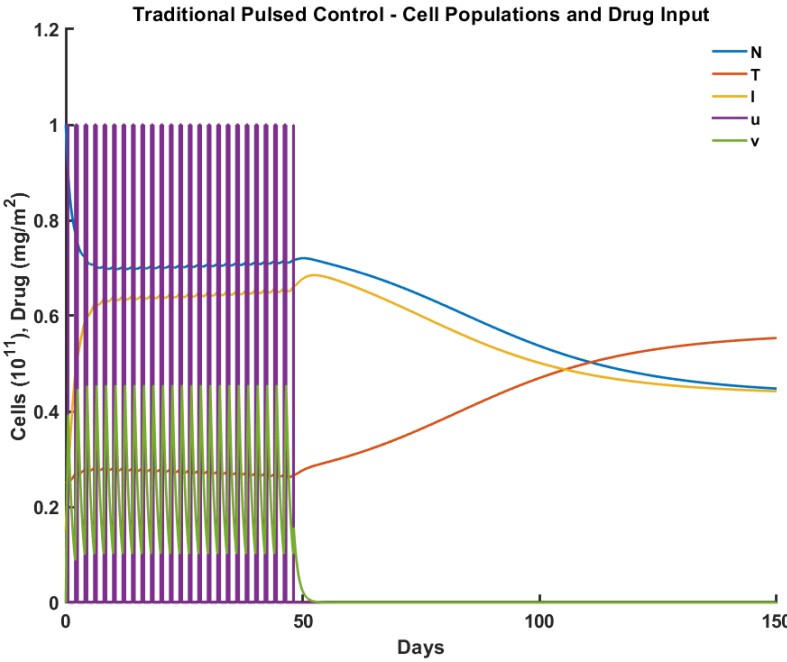

**Figure 11.** Pulsed control therapy. Individual cell populations evolution under the computation of the robust control law of drug dose $v$. The pulsed control computes densely timed, equally sized, and dosed drug administrations which only trigger short decays in tumor (T) proliferation, which at the end of the therapy resumes its uncontrolled growth which suppresses the normal cells (N) production and a weakening immune response (I).

Our study departs from the traditional control problem formulation and goes beyond the analysis of the tumor-immune-drug model in order to have a more complete understanding of the implications of the existence of basins of attraction in the treatment process and design a therapy control scheme that goes beyond robustness and reaches antifragility, as we described in the previous section.

## 3. Experiments

To evaluate the antifragile control synthesis, we simulated our tumor-immune-drug network model without and under the impact of drug administration. For the simulation, we used the Matlab environment on a standard PC. In order to implement the robust and optimal control approaches, we used the library developed by Kelly et al. [51]. The model parametrization of the tumor-immune-drug model is done as given by dePillis et al. [12,13]. The full list of parameters of the tumor-immune-drug network model is given below in Table 1.

**Table 1.** Tumor–immune–drug network model parameters.

| Model Parameter | Parameter (Initial) Value | Impact on Dynamics |
|---|---|---|
| Number of normal host cells $N$ | 1.0 | Interacts with $I$ and $T$ w/wo $u$ |
| Number of tumor cells $T$ | 0.25 | Interacts with $N$ and $I$ w/wo $u$ |
| Number of host immune cells $I$ | 0.1 | Interacts with $N$ and $T$ w/wo $u$ |
| Drug concentration at the tumor location $u$ | 0.01 | induces cell kill by toxicity (normal+tumor) |
| Drug administration $v$ | 0.0 | modulates the drug concentration at the tumor |
| Fraction normal cell kill $a_1$ | 0.2 | dose-related weight of normal cell kill (toxicity) |
| Fraction tumor cell kill $a_2$ | 0.3 | dose-related weight of tumor cell kill |
| Fraction immune cell kill $a_3$ | 0.1 | dose-related weight of immune cell kill |
| Tumor growth rate $r_1$ | 1.5 | typically $c_2 + c_3$ |
| Normal cells growth rate $r_2$ | 1.0 | per capita growth |
| Carrying capacity of tumor cells $b_1$ | 1.0 | weights tumor cells self-excitation |
| Carrying capacity of normal cells $b_2$ | 1.0 | weights normal cells self-excitation |
| Tumor-immune competition factor $c_1$ | 1.0 | competition |
| Immune-tumor competition factor $c_2$ | 0.5 | modulation |
| Tumor-normal competition factor $c_3$ | 1.0 | competition |
| Normal-tumor competition factor $c_4$ | 1.0 | competition |
| Immune cells death rate $d_1$ | 0.2 | regulation through $\frac{s}{d_1}$ |
| Drug influx modulation $d_2$ | 1.0 | rate of change/decay of $u$ |
| Immune threshold rate $\alpha$ | 0.3 | related to the immune response curve |
| Immune response rate $\rho$ | 0.01–1.0 | immune-compromised to healthy |

To evaluate the therapy controllers, we considered metrics of interest for the clinical use practice over a typical 150 days of chemotherapy. The overall evaluation is given in Table 2. The time to tumor elimination describes the number of days until the tumor decreased under a threshold of $0.05 \times 10^{11}$ cells. For the drug impact assessment, we defined the maximum drug concentration within the patient's body over the therapy duration and the total drug administered over the entire therapy duration. Finally, we also considered the impact the therapy has on the tumor and the normal cells. This is characterized by the maximum number of tumor cells and the minimum number of normal cells, which describe the maximum size of the tumor during the therapy and the highest kill of normal cells during the therapy, respectively.

**Table 2.** Therapy control algorithms evaluation: evaluation criteria and candidate approaches.

| Criteria/Control | Pulsed | Robust | Optimal | Antifragile |
|---|---|---|---|---|
| Time to tumor elimination (days) | 150+ | 68 | 100 | **65** |
| Max drug concentration (mg/L) | **0.455** | 0.987 | 0.659 | 0.736 |
| Total drug administered (mg/L) | 15.00 | **12.00** | 15.00 | 14.31 |
| Max tumor cells ($\times 10^{11}$ cells ) | 0.553 | 0.258 | 0.257 | **0.251** |
| Min normal cells ($\times 10^{11}$ cells) | 0.447 | 0.706 | **0.750** | 0.592 |

The pulsed control is a standard approach for chemotherapy that uses a train of impulses to deliver the drug. The parametrization of this type of control assumes the choice of the number of impulses, the inter-impulse time interval, and the drug dose administered through each impulse. For our experiments, we considered 25 impulses of amplitude 1 administered over the first 50 days of therapy (see Figure 11).

The robust control implementation was based on the computation of the control Hamiltonian of the optimal control problem and parametrized according to the work of Ledzewicz et al. [47]. The drug dosage was restricted to a maximum dose of 1 mg/L but with a varying administration, time intertwined with no drug administration for an overall 18 effective administration days of the 150 total therapy days, as shown in Figure 10.

The optimal control problem was formulated based on an objective function that weighted the tumor size evolution, the total tumor size, the maximum tumor size along the therapy, and the drug concentration—see Equation (29). The optimal dosing was irregular in both amplitude and duration, determining a rather fast decrease of the overall concentration in the first days of therapy, as depicted in Figure 9.

Finally, for the antifragile control the drug dosing was computed such that the survival curve (i.e., tumor size vs. drug dose) points towards the antifragile region, that is the convex region of the curvature of the survival curve (i.e., the second derivative of Hill function). The antifragile control law, composed of the PD-like anticipation component and the SMC-like variable structure component, pushes the dynamics of the tumor-immune-drug model to a state where the drug concentration ensures a fast elimination of the tumor—see Figure 8. For the detailed parametrization of the simulations please check the source code available on GitLab (Code available at: https://gitlab.com/akii-microlab/antifragile-therapy-ctrl, accessed on 31 August 2022).

## 4. Discussion

In clinical practice, it is typical to follow a set treatment regimen in which a steady dosage is given regularly. There are, however, a large number of noteworthy cases where the tumor responds better to a "volatile" treatment plan than continuous therapy. Such a volatile (or synonymously an "uneven") dose control can actually push the tumor growth dynamics towards instability and, under the joint effect of the drug and immune system, towards elimination. We explored these cases in our experiments using asymmetric networks of the tumor-immune-drug interactions.

In our study, we were interested to explore how antifragile control can offer a systematic approach to controlling drug dosage in "volatile" treatment plans where the drug "unevenness" can determine a fast tumor elimination with a good trade-off in drug quantity administration and collateral normal cell damage (i.e., toxicity induced cell death). We summarize our findings in Table 2. Here we notice how each control approach determines gains in one of the multiple dimensions of the therapy regimen efficiency. When considering time to tumor elimination, the antifragile controller is able to reduce the tumor under the threshold of $0.05 \times 10^{11}$ cells in under 65 days whereas the robust control reaches the same tumor burden in 68 days, as depicted in Figure 10—red trace. This demonstrates that the antifragile control can compensate for the anti-symmetric interactions in the tumor-immune-drug model and push the system towards a faster tumor reduction (see therapy day 65 in Figure 8).

Considering the drug administration, the traditional pulsed approach stands out, with the lowest maximum drug concentration among all methods. This comes also with the price of having just a limited impact on the tumor growth dynamics, which starts to diverge once the regimen reaches the maximum delivery program, around day 52 in Figure 11. All the other methods determine a higher maximum drug concentration, with up to 80% more, but with the better outcome of reducing the tumor burden in between 65 up to 80 days—see Figures 8–10. Interestingly, the most drug quantity efficient approach is the robust control where only 12 mg/L were administered during the therapy, but due to its delivery pattern (i.e., frequency and timing) reached the highest maximum drug concentration of

0.987 mg/L. The antifragile control, utilized a moderate total drug dose of 14.31 mg/L distributed over the entire 150 days of therapy for a maximum dose of 0.736 mg/L, as seen in Figure 8.

Considering the individual evolution of the cell populations in the tumor-immune-drug model, we can see that the pulsed therapy is the one that determined the largest tumor size of $0.553 \times 10^{11}$ cells at day 150 of the therapy regimen, due to its limited delivery schedule and the conservative dosage. This also determines its inefficiency in tumor elimination and, of course, a high loss in the preservation of the normal cells, of which many were killed due to a high toxicity level. This is visible through the large loss of normal cells of $0.447 \times 10^{11}$ cells at the end of therapy. On the other side of the spectrum, the highest tumor kill strength was produced by the antifragile control, with close results by the robust and the optimal control approaches. This is, of course, motivated by the weighted objective functions in the case of optimal and robust controllers, and the advantages of the anticipation and the high-frequency control activity of the antifragile control. With a minimal tumor burden of $0.251 \times 10^{11}$, the antifragile controller is the one which elicited a sustained, high-frequency control activity for the entire duration of the therapy, but with moderate drug administration and concentration—see Figure 8.

When considering the therapy damage to normal cells, quantified by the minimum number of normal cells, the optimal control stands out with a maximum of $0.750 \times 10^{11}$ cells, followed by the robust and antifragile approaches. Overall, the antifragile control provides a fast reduction of the tumor burden, given a moderate drug administration and drug concentration, with the highest tumor kill and moderate collateral damage of the normal cells through drug induced toxicity.

From a quantitative point of view, we see in Table 2, that each of the advanced control methods (i.e., robust, optimal, and antifragile) excel at one of the clinically relevant criteria. Yet, there is a more subtle comparison to be made from the point of view of their implementation details and their generic behavior. First, a main difference between robust control and antifragile control refers to the sensitivity to uncertainty and volatility (here used as evenness vs. unevenness of the dosage). Robust control excels in handling (up to a point) both structured (i.e., noise, parameter deviations) and unstructured (i.e., unmodelled dynamics) uncertainties with the price of an irregular timing of the on-off control signal. On the other size, antifragile control has a positive sensitivity to dispersion in model parameters and dynamics, especially when the irregular control signal pushes the system trajectory to the desired (antifragile) region of the reference (dosage). Second, robust control can handle the intrinsic (i.e., model parameters distribution) and inherent (i.e., exogenous perturbation of the immune system) fragility of the control loop, whereas antifragile control implements the transfer function towards induced antifragility (see Taleb et al., 2012 [52] for mathematical proofs). In other words, antifragile control goes beyond the capabilities of robust control through: (1) inducing a "convex exposure", and hence a volatility gain, when driving the system trajectory to the antifragile region of the drug response, and (2) limit the "concave exposure" through the damped control of dosage when reaching fragility on the drug response curve. The closed-loop system actually gains (i.e., faster tumor kill) from the covariance pattern of the joint tumor-immune-drug dynamics—see Figure 8).

Our study demonstrates that through the antifragile control framework we can find a trade-off between the time to reduce the tumor burden and the drug administration. This happens while maximizing the tumor kill and conserving the normal cells given the complex anti-symmetric interactions among the tumor, the immune system, and the chemotherapeutic agent. Although this is an initial instantiation of antifragile control, we believe that such a control framework can gain from the uncertainty and volatility describing the system's dynamics. This can be achieved by building a strong anticipation component and a versatile structure change driving the closed-loop system towards the reference behavior.

Supported by previous studies, we foresee translation and validation in clinical practice. The initial experiments will be done on cell lines (see the multiple studies of

Axenie et al. [3,45,53]), which simultaneously determine the considered schedule and the dose. The next step is to extract the dose-response curve from the assays of the considered cell line under therapy schedules with clinically relevant chemotherapeutic agents. As cell lines are individually resistant to therapies a cross-sensitivity analysis will be needed to confirm the dose-response curve across the considered cell lines. The extracted dose-response curve will quantify fragile and antifragile regions for both treatment resistance and collateral sensitivity, as shown in West et al. [5]. The static dose-response curve will be then used as a desired dynamics in the control system which will generate a corresponding dose value to push the closed-loop system (i.e., drug dose–immune system–tumor) towards the antifragile region (i.e., high efficacy, low toxicity).

We must note that this preliminary study is based on some design considerations which are supported by literature, such as the studies of Amin et al. [54] and Grommes et al. [55], but not fully validated. For instance, experimentally validating the hypothesis that links the shape of the dose-response curve and treatment schedule is already evident in recent studies such as the one of Chmielecki et al. [56] or the in-vivo study of Schöttle et al. [57].

Finally, despite the attractive properties it has, the system faces at the moment a set of challenges which, we believe, might affect its translation to oncological practice. First, we still need multiple in-vitro and in-vivo studies to validate the (desired) drug response curve patterns, especially when considering the pharmacokinetics and pharmacodynamics of the various compounds used depending on the type(or sub-type) of the tumor, as shown in Chmielecki et al. [56], and Schöttle et al. [57]. Second, we need to reach a consensus on the "evenness or unevenness" of the drug schedules in therapy as the efficacy of the proposed control system depends on the degree of drug resistance in the tumor, and thus changes over the course of therapy, as developed in Amin et al. [54], and West et al. [5]. Third, there is still a strong belief in static algorithms built by national oncology forums, where, based on studies, therapy schemes are designed based on a decision tree of patient-specific parameters. Such schemes have no adaptive component (as opposed to our approach), they simply try to capture all possible variable constellations. We believe that our approach supports the departure from the "traditional MTD" (maximum tolerated dosage) towards targeted therapy, where therapy efficacy plateaus faster as the growth of therapy side-effects.

## 5. Conclusions

Cancer therapy control is a complex intervention that revolves around understanding the complex and intricate interactions among the tumor, the immune system, and the chemotherapeutic drug and breaking the tie towards minimal tumor burden. Aside from the traditional pulsed therapy schemes, multiple control-theoretic approaches, such as robust and optimal control, were developed to capture the complex tumor-immune-drug dynamics and compute an optimal and robust drug dosing that minimizes the tumor and preserves as much as possible normal cells. However, there is always a trade-off between tumor reduction and normal cell kills. Antifragile control emerges as a control-theoretic framework capable of gaining from: (1) the uncertainty describing tumor-immune-drug interactions, (2) the volatility of drug response curves in patient populations, and (3) the variability in tumor response, immune capabilities, and drug resistance patterns of a patient-tumor pair. This is achieved through the formal implementation in a control-theoretic framework of the three main pillars: (1) redundant overcompensation, (2) structure variability, and (3) high-frequency control activity. These three ingredients are concretely implemented in a control structure cast in the Riemannian geometry where a PD control-like structure is combined with an SMC control-like component to elicit strong anticipation and convergent dynamics. As our experiments emphasize, this novel control-theoretic approach has the potential to go beyond the robust and optimal control with the advantage that the closed-loop system doesn't need to model uncertainty, adapt to changes, or seek optimality. Antifragile control anticipates future variability in system dynamics and exploits drug dosing volatility for a fast tumor burden reduction with a trade-off in drug concentration and collateral normal cell kill, through the induced drug toxicity. Our next steps will

focus on a broader formalization of the antifragile control theory and its instantiation in a data-driven control loop, where the tumor-immune-drug dynamics are learned from data and not given by a parametric model. This assumes simultaneously learning tumor growth curves, pharmacokinetics, and immune responses. We are confident that this approach will make antifragile control more versatile and our initial results in the present study and the one of Kurz et al. [3] are encouraging. Finally, we believe that the antifragile control of non-parametric system models can be highly relevant, as it could additionally capture aspects such as drug resistance, specific immune cell types dynamics, and the incorporation of the spatial heterogeneity of tumors. We believe that such an approach will strengthen oncologists' therapy tool-set by informing therapy design and patient tailoring.

**Author Contributions:** C.A. designed the research, designed the framework, designed the experiments, implemented simulation software, analyzed and interpreted the data and results, D.K. designed the experiments, analyzed and interpreted the data and results, M.S. designed the framework, implemented simulation software, analyzed and interpreted the data and results. C.A., D.K. and M.S. wrote and revised the manuscript. All authors have read and agreed to the published version of the manuscript.

**Funding:** This research received no external funding.

**Institutional Review Board Statement:** Not applicable.

**Informed Consent Statement:** Not applicable.

**Data Availability Statement:** Not applicable.

**Conflicts of Interest:** The authors declare no conflict of interest.

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
