# Peer review of "Antifragile Control Systems: The Case of an Anti-Symmetric Network Model of the Tumor-Immune-Drug Interactions"

_symmetry, doi:10.3390/sym14102034_

Round 1
Reviewer 1 Report
The manuscript is well drafted and would be a valuable publication.
Since the manuscript contains several technical terms which may be not familiar to medical faculty, it is vital to provide sufficient context through descriptions. Also, any dry lab must be backed by a wet lab. Thus, at the least provide a blue print on how to validate the findings of this study through experiments research beyond computational biology. Provide a brief on the current challenges in translating the concept on to clinical practice
Author Response
I would like to thank Reviewer 1, on behalf of the authors, for the very good points raised in the review suggestions.
The manuscript is destined for the technical community, to assess the problem formulation, the mathematical apparatus, and the potential the approach has (in simulation) to contribute to drug dosing in chemotherapy. But, your great suggestions were addressed and, now, the manuscript has been strengthened to capture the principles and challenges for clinical translation.
I will address the points one by one by referring to the lines in the new revision where the supporting context descriptions were added.
- "Since the manuscript contains several technical terms which may be not familiar to medical faculty, it is vital to provide sufficient context through descriptions."
- we have complemented the technical (computational biology, control theory jargon with contextual descriptions of functionality and impact for practice. We invite the reviewers to consult in the revised manuscript version the newly added/updated text in blue in the Introduction section along with the newly added Figure 1.
- "Also, any dry lab must be backed by a wet lab. Thus, at the least provide a blueprint on how to validate the findings of this study through experiments and research beyond computational biology."
- although the study focuses on the fundamental understanding of how can such a control-theoretic approach can drive drug-dosing into high efficacy regions of the drug response curve, we also added some ideas on how can we validate the approach. We invite the reviewers to consult in the revised manuscript version the newly added/updated text in blue in the Discussion section.
- "Provide a brief on the current challenges in translating the concept onto clinical practice."
- we have identified three initial challenges we foresee in the clinical translation of the system, supported also by parallel studies in the community.
- We invite the reviewers to consult in the revised manuscript version the newly added/updated text in blue in the Discussion section.
Reviewer 2 Report
In this manuscript, Axenie et al. described a new computational model, an antifragile control system, that considered the dynamics of tumor growth, the immune response and the drug dosage toward optimizing dosing and treatment interval for cancer in general. In addition, the authors compared the performance of antifragile system with that of other control systems, including pulsed, robust, and optimal through MATLAB simulation. They found that the new antifragile system is very effectively in shrinking the size of the tumor within 65 days, especially compared to pulsed and optimal system, which require over 100 days. While this topic is of significance and has a far reaching impact, a few minor issues need to be addressed prior to publication:
1. On line 138, further explanation is needed on why 10^11 cells were picked as the reasonable scale. Either a citation or a short explanation of what size of the tumor 10^11 cells corresponds to would help justify this assumption. Further, all tumor sizes discussed within this manuscript have less than 10^11 cells. Therefore, it would make more sense to assume a tumor has 0.5*10^11 cells?
2. From a clinical perspective, using an efficient quantity of drugs, shrinking the size of the tumors relatively quickly and also having a small total loss of normal cells are all advantages that the robust control system has. A more detailed discussion on why the antifragile system is better than the robust control system is necessary to further highlight the significance of this model.
Author Response
I would like to thank Reviewer 2, on behalf of the authors, for reading the manuscript and for the positive feedback. We have now addressed all the raised points and we invite the reviewer to check the newly added/updated text marked in the revised version with the colour magenta and index R2.
- We have added pointers to studies supporting the parametrization of the tumour population in our model. We have updated the initial values accordingly.
- We have added in the Discussion section some experiments-supported statements on the advantages of antifragile control over robust control.
Reviewer 3 Report
The manuscript is novel and quite comprehensive. Few minor revisions are required as mentioned in the pdf

Author Response
I would like to thank the reviewer, on behalf of all authors, for the time and the very good suggestions to improve the manuscript.
We have addressed all the comments in the current revision and newly added/updated text in the revision is marked by the colour orange and index R3. Additionally, I have updated the citation type to contain the name of the author and the reference index as suggested.
Round 2
Reviewer 1 Report
Manuscript can be accepted